# Effect of multiple risk behaviours in adolescence on educational attainment at age 16 years: a UK birth cohort study

Caroline Wright, Ruth Kipping, Matthew Hickman, Rona Campbell, Jon Heron

Population Health Sciences, Bristol Medical School, University of Bristol, Bristol, UK

**Correspondence to**
Dr Caroline Wright;
caroline.wright@bristol.ac.uk

## ABSTRACT

**Objectives** To explore the association between adolescent multiple risk behaviours (MRBs) and educational attainment.

**Design** Prospective population-based UK birth cohort study.

**Setting** Avon Longitudinal Study of Parents and Children (ALSPAC), a UK birth cohort of children born in 1991–1992.

**Participants** Data on some or all MRB measures were available for 5401 ALSPAC participants who attended a clinic at age 15 years and/or completed a detailed questionnaire at age 16 years. Multiple imputation was used to account for missing data.

**Primary outcome measures** Capped General Certificate of Secondary Education (GCSE) score and odds of attaining five or more GCSE examinations at grades A*–C. Both outcome measures come from the National Pupil Database and were linked to the ALSPAC data.

**Results** Engagement in MRB was strongly associated with poorer educational attainment. Each additional risk equated to −6.31 (95% CI −7.03 to −5.58, p<0.001) in capped GCSE score, equivalent to a one grade reduction or reduced odds of attaining five or more A*–C grades of 23% (OR 0.77, 95% CI 0.74 to 0.81, p<0.001). The average cohort member engaged in 3.24 MRB and therefore have an associated reduction in GCSE score equivalent to three and a half grades in one examination, or reduced odds of attaining five or more A*–C grades of 75%.

**Conclusion** Engagement in adolescent MRB is strongly associated with poorer educational attainment at 16 years. Preventing MRB could improve educational attainment and thereby directly and indirectly improve longer-term health.

## Strengths and limitations of this study

► This is the first longitudinal study in the UK to examine the association between multiple health risk behaviours and educational attainment.
► Owing to the scale and richness of the Avon Longitudinal Study of Parents and Children data, we have been able to control for a wide range of confounders, including socioeconomic measures, previous educational attainment and IQ.
► The risk behaviours were all reduced to binary variables in order to construct the total multiple risk behaviour (MRB) score, which leads to a loss of information.
► By summing the risk behaviours to create a measure of total MRBs, we attribute equal weight to each behaviour, however, we found no strong evidence for an alternative classification of MRB in terms of clusters of specific MRBs or latent classes.
► There is considerable missing data for the exposure and confounder variables which reduces power and may introduce bias, however, there is no missing data on either outcome measure, and although missingness is related to MRB, our imputed analyses are similar to the complete case analysis.

## INTRODUCTION

Health risk behaviours such as smoking, alcohol consumption, physical inactivity and unhealthy eating are prevalent during adolescence[1 2] and have also been shown to co-occur during this period.[3–5] A growing body of evidence suggests that these behaviours are strongly associated, some causally, with adverse health outcomes in later life, including chronic health conditions, morbidity and premature mortality.[5–7] Further, evidence has shown that multiple risk behaviours (MRBs) are cumulatively associated with cardiovascular and all-cause mortality.[5 6] For example, having four healthy lifestyle behaviours is associated with a fourfold difference in mortality compared with having none.[6] Similarly, a composite measure of MRB predicted preventable death, over and above the predictive value of single lifestyle behaviours.[8] It has been posited that many lifestyle behaviours are underpinned by the same neural circuitry, and as such when intervening on one behaviour, changes in other behaviours can be expected.[9] Finally, interventions responding to multiple risks may be more efficient and better value for money, providing potentially huge benefits for considering multiple, rather than single risk behaviours.

Successful completion of compulsory education is important to an individual's well-being and lifelong opportunities. Those with lower educational attainment are more



likely to smoke,[10] be overweight, and have poor physical and mental health outcomes.[11] They also experience reduced employment opportunities and earning potential.[11–13] Successful completion of compulsory education is strongly associated with increased aspirations and life satisfaction[14] and those with college degrees or higher are the most likely to engage in healthy behaviours.[15]

Many studies consider the effects of single health risk behaviours and educational outcomes. Obesity,[3 16] smoking tobacco,[4 17 18] using cannabis,[17–19] drinking alcohol,[4 18] self-harm,[20] physical inactivity[21] and screen-based behaviours involving television, internet or computer games[22] are all associated with poorer educational outcomes. Less studies consider MRBs simultaneously,[1 23–25] fewer still consider a large number of heterogeneous risk behaviours concurrently.[26–28] Past research has shown that those with no or intermediate qualifications are more likely to engage in MRB compared with those who attended higher education.[15 29–31] However, these studies relate to adult populations and have considered only a limited number or range of risk behaviours. No UK studies, to our knowledge, have examined engagement in MRBs and educational outcomes. Using data from the Avon Longitudinal Study of Parents and Children (ALSPAC) cohort in England, we aimed to investigate the association of MRBs during adolescence and educational attainment at age 16.

## METHODS
### Sample
Data were drawn from the ALSPAC, an ongoing prospective observational population-based study investigating the effects of a wide range of influences on health and development across the life course. Pregnant women residing in the old administrative county of Avon, who had an estimated date of delivery between 1 April 1991 and 31 December 1992, were invited to participate. The initial study cohort consisted of 14 062 live-born children of whom 13 988 singletons or twins were still alive at 12 months of age. A small number of participants withdrew from the study (n=24).[32] Those who were neither enrolled nor part of the original core ALSPAC sample were excluded from our analyses, along with any triplets or quadruplets whose identity, because of their rarity, would be compromised. As Boyd *et al* observe the ALSPAC 'enrolled sample' are more likely to be white (OR 3.85 (95% CI 3.50 to 4.24), p<0.001) and less likely to be eligible for free school meals (FSMs) (OR 0.46 (95% CI 0.43 to 0.50), p<0.001) than the National Pupil Database (NPD) key stage 4 (KS4) government-maintained establishments national sample[i] and 'recent responders' to ALSPAC are more likely to be female (OR 1.88 (95% CI 1.74 to 2.03), p<0.001), white (OR 1.34 (95% CI 1.10 to 1.62),

p=0.004) and less likely to be eligible for FSMs (OR 0.51 (95% CI 0.44 to 0.60), p<0.001).[32] The study website contains details of all the data that are available through a fully searchable data dictionary: http://www.bris.ac.uk/alspac/researchers/data-access/data-dictionary/.[33]

### Linkage between ALSPAC and NPD
The NPD is a pupil-level data source which matches pupil and school characteristic data to pupil-level attainment data in England. The Fischer Trust completed the linkage between the NPD and ALSPAC data in 2002. It is only mandatory for schools following the national curriculum to contribute to the NPD. Independent schools may provide attainment data on a voluntary basis.

### Exposure measure
#### MRBs at age 16
Measures of participation in 13 distinct risk behaviours at the ages of 15 and 16 years were derived from participants' responses at two ALSPAC data collections during their late teens (see table 1). The first was a self-completed questionnaire issued during a clinic attended at age 15 (median age 15 years and 5 months) and the second comprised responses to a postal questionnaire administered at age 16 (median age 16 years and 7 months). The MRB measure was informed by the work of Hurrelmann and Richter, who present an integrative model of risk behaviour. They argue that while inadequate coping processes are ubiquitous during adolescence, these processes can result in very different health risk behaviours among young people. However, despite variations in presentation (eg, physically hurting someone on purpose, vs not wearing a seat belt), risk behaviours reflect the very similar dimensions of either externalising, internalising or evasive forms.[34] Lending weight to this position, previous analyses using this measure of MRB have shown that health risk behaviours are patterned according to gender. For example, antisocial and criminal behaviours, cannabis use and vehicle-related risk behaviours are more prevalent among males, while tobacco smoking, self-harm and physical inactivity are more prevalent among females. However, despite the gendered patterning of single risk behaviours, females and males engaged in a similar number of risk behaviours.[35] Similarly, another previous analysis showed that while the associations between individual risk behaviours and measures of socioeconomic status (parental social class, maternal education and income quintile) were highly variable, a more consistent relationship was established between the MRB measure and socioeconomic status. When compared with the highest social class, maternal education or income quintile, the odds of engaging in a greater number of MRBs increased for each incremental decrease in social position.[23] Finally, an analysis of these MRB data, using latent class analysis,[36] showed that the resulting classes simply varied according to the number of risk behaviours, rather than demonstrating distinct risk profiles based on classes of behaviours. Having found no strong evidence for

---

[i] Refers to all pupils, excluding those in ALSPAC, from English government-maintained establishments (GMEs) who sat their KS4 assessments during the same academic years as the ALSPAC cohort (academic years 2007–2009).

**Table 1** Multiple risk behaviours and their derivation

| Health risk behaviour | Definition/how derived |
| --- | --- |
| Physical inactivity | Young person (YP) has typically over the past year exercised <5 times per week. |
| TV viewing | YP spent 3 or more hours watching TV on average per day across the week. |
| Car passenger risk | YP had been in a car passenger at least once in their lifetime where the driver (1) had consumed alcohol or (2) did not have a valid licence, or (3) the YP chose not to wear a seat belt last time travelled in a car, van or taxi. |
| Cycle helmet use | If the YP reported that they had last ridden a bicycle within the previous 4 weeks and they had not worn a helmet on the most recent occasion. |
| Scooter risk | YP has driven a motorbike/scooter off road, or without a licence on a public road at least once. |
| Criminal/antisocial behaviour | YP reported that at least once in the past year they had undertaken at least one of the following seven offences: carried a weapon; physically hurt someone on purpose; stolen something; sold illicit substances to another person; damaged property belonging to someone else either by using graffiti, setting fire to it or destroying or damaging it in another fashion; subjected someone to verbal or physical racial abuse; or been rude/rowdy in a public place. |
| Hazardous alcohol consumption | In the past year had scored 8 or more on the Alcohol Use Disorders Identification Test indicating hazardous alcohol consumption. |
| Regular tobacco smoking | Has ever smoked and is regularly smoking by currently smoking at least one cigarette per week. |
| Cannabis use | Those who reported using cannabis 'sometimes but less often than once a week' or more regular use were classified as occasional users. |
| Illicit drug/solvent use | In the year since their 15th birthday, YP had either been a regular user (ie, used five or more times) of one or more illicit drugs (excluding cannabis) including amphetamines, ecstasy, lysergic acid diethylamide (LSD), cocaine, ketamine or inhalants including aerosols, gas, solvents and poppers. |
| Self-harm | Young people who said they had purposely hurt themselves in some way in their lifetime. |
| Penetrative sex before age 16 | YP reported having had penetrative sex in the preceding year and that they were under 16 at the time. |
| Unprotected sex | Penetrative sex without the use of contraception on the last occasion they had had sex in the past year. |

Sources of information:

Age 15 years clinic: criminal and antisocial behaviour, penetrative sex prior to age 16, and unprotected sex.

Age 16 years questionnaire: physical inactivity, TV viewing, car passenger risk, cycle helmet use, scooter risk, hazardous alcohol drinking, regular smoking, illicit drug/solvent use and self-harm.

TV, television.

employing an alternative classification of MRB based on classes of behaviour, the MRB measure comprises a count of the number of risk behaviours representing a breadth of domains of social and health risk including: sexual health, substance use, self-harm, vehicle-related injury risk, criminal and antisocial behaviour, and physical inactivity. The derivation of each behaviour is discussed in more detail in an earlier paper.[23] For the purposes of the analyses reported here a total number of risk-behaviours score from 0 to 13 was derived for each participant.

### Outcome measures
#### Educational attainment at age 16

Pupils in England aged between 14 and 16 years complete compulsory schooling during school years 10 and 11 and take their General Certificate of Secondary Education (GCSE) (or equivalent) examinations, this is referred to as KS4. At the time that the ALSPAC cohort were in school, UK law stated that pupils were to remain in compulsory education until the age of 16, so unlike A-levels, which are taken 2 years later and are optional, GCSEs are one

of very few occasions in a young person's life when their educational attainment is assessed along with most of their peers. Two outcomes relating to KS4 educational attainment were used in the analysis. Achieving five or more A*–C grades at GCSE was chosen because it is a minimum requirement for many post-16 education and training courses and as such represents an important threshold for young people to exceed. The second outcome takes the individual scores for each GCSE, which are calculated as A*=58 through to G=16 and ungraded U=0 (unlike in North America where grades range from A to F). This general attainment score is calculated by summing a pupil's eight best grades, referred to as the capped GCSE score. It is seen as preferable to a total GCSE score because it represents the same measure that is used in the published value-added school league tables, which have become an important measure of the quality of education provision. It is also considered fairer than the total (uncapped) score since it moderates the scores of pupils who score highly merely by taking more examinations.

## Possible confounders

We adjusted for a number of known confounders: sex, season of birth, parent's highest social class (professional; managerial and technical; skilled non-manual; and skilled manual, part or unskilled manual), mother's highest educational level (degree, A-level, O-level/GCSE and less than O-level/GCSE), household income (divided into quintiles of high to low income), housing tenure during pregnancy (mortgaged or own property, privately rented property or subsidised rental property) and claiming eligibility for an FSM. Season of birth has been shown to be an important predictor of educational attainment. In England, where the academic year runs from 1 September to 31 August, children who are born in the autumn tend to outperform those who are born in the summer.[37] We additionally controlled for IQ score at age 8 years and key stage 2[ii] educational attainment in order to reduce the likelihood of reverse causality between early educational performance and engagement in MRBs. Analysis of confounders and both the exposure and outcomes variables was conducted and can be found in the online supplementary material.

## Missing data

Of the starting sample of 13 954 subjects (enrolled cohort, singletons and twins alive at 1 year): 2618 (18.8%) did not have a linked education record for KS4 and were excluded from the analysis on that basis. There are a number of possible reasons for this type of missingness. Participants may not have the linked data from the NPD, the participant may have withheld consent, or the participant may have been attending a school that does not follow the national curriculum, that is, an independent school. Independent school education is of particular interest in this case because of its prominence in Bristol and Avon. Between 2006 and 2009 (when the ALSPAC cohort would have taken their GCSE examinations), the percentage of pupils educated in independent schools in England remained stable at approximately 7%. In Bristol, it ranged between 15.1% in 2006/2007 and 13.4% in 2008/2009. However, with no way of confirming that those with missing attainment and school type data were independently educated, and no alternative identifier of independent school status, we were unable to conduct a sensitivity analysis with this respect. Our sample analysis is therefore less representative of Bristol at the time, but more generalisable to the overall population, where independent schooling is less common. Of the 11 336 subjects with education outcome data, 8398 (74.08%) were invited to the clinic and of those, 4534 (53.99%) attended and 8017 (70.72%) were sent the questionnaire and of those, data were available for 4052 (50.54%). Overall,

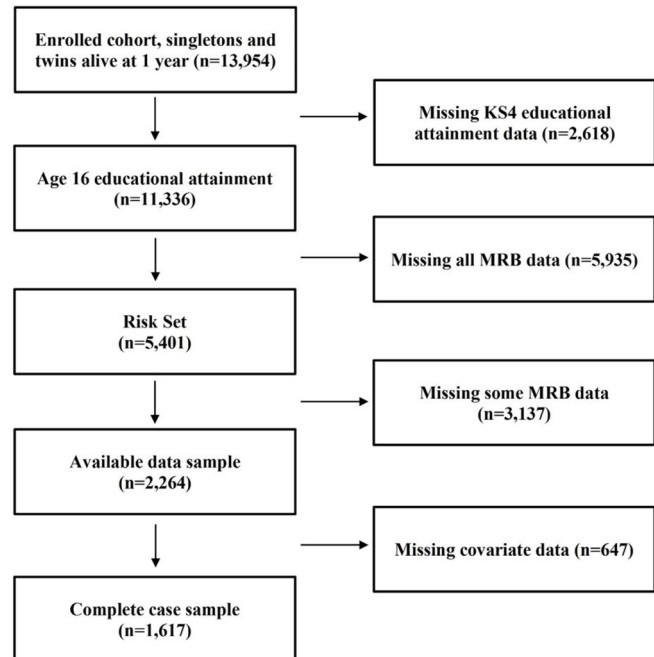

**Figure 1** Deriving the sample. KS4, key stage 4; MRB, multiple risk behaviour.

5401 participants had some or all of their MRB information and this became the imputation sample on which our analysis is based. Additional analysis regarding those with complete data (n=1617) is available in the online supplementary material (see figure 1 for how we derived the sample).

Multivariate imputation by chained equations was carried out using the 'ice' routine in Stata V.14. This approach is based on the missing at random assumption, that is, that any differences between the missing and observed values, can be explained by differences in the observed data.[38] All variables used in the analyses including all measures of MRB, educational attainment and potential confounders were included in the imputation model, along with many other measures of the exposures, outcomes and confounders that had been collected at different time points. These are included in the imputation routine as auxiliary variables to reduce bias by improving the precision of the imputation model. Monte Carlo errors were used to compare the results obtained when imputing 25, 100, 250 and 500 data sets.[39] Imputed results shown have been pooled across the 500 data sets. Among this sample, there was an average missingness of 18.23%.

## Patient and public involvement

No patients were involved in the design of this study. However, our choice of risk behaviours was informed by discussions with two groups of adolescents through the Centre for the Development and Evaluation of Complex Interventions for Public Health Improvement (DECIPHer) Advice leading to Public Health Advancement (ALPHA) young person's research advisory

---

[ii] Key stage 2 is the legal term which refers to the 4 years of schooling in maintained schools in England and Wales when pupils are aged between 7 and 11 years. Key stage 2 culminates in standardised assessment tests (SATS) at age ~11 years, the results of which have been used here.

group (http://decipher.uk.net/public-involvement/young-people/).

## Sample descriptive statistics

Compared with the imputation sample, the complete case sample had better educational outcomes, a higher mean IQ score and a lower mean total MRB score. The percentage of females, and all indicators of advantageous socioeconomic position were over-represented in the complete case sample when compared with the imputation sample (see table 2).

## Associations between confounders and exposure variables

We found that lower maternal education, lower parental social class, living in privately rented or subsidised housing, having a lower income and claiming eligibility for FSMs were all positively associated with the total number of MRBs at age 16 years. IQ at age 8 years and higher previous educational attainment at key stage 2 were negatively associated with total MRB score. There was no association between sex, season of birth or special educational needs and MRB score.

Lower maternal education, lower parental social class, living in privately rented or subsidised housing, having a lower income and claiming eligibility for FSMs were all associated with lower educational attainment at age 16 years. This was true for both educational outcomes. Being born in the spring or summer compared with the autumn was associated with lower capped GCSE score, but had no association with gaining five or more A*–C grades at GCSE. Having higher previous attainment levels at key stage 2 and a higher IQ at age 8 years were associated with better educational outcomes at age 16 years and being female was associated with an improved grade point average equivalent to more than one and a half grades. Details of these analyses can be found in the online supplementary material.

## Statistical analysis

Linear regression was used to examine associations between total MRB and the continuous outcome, capped GCSE score and logistic regression was used for the binary outcome, gaining five or more GCSE grades A*–C. Analyses were conducted on the complete case and imputed data the latter of which constitutes the main analysis. We ran unadjusted models for both outcomes followed by a sequence of models adjusted for: (1) sex and season of birth, (2) sex, season of birth, maternal education, parental social class, household income, housing tenure and FSMs, and (3) sex, season of birth, maternal education, parental social class, household income, housing tenure, FSM, IQ and previous educational attainment. We tested for non-linearity between exposure and outcome using the 'nlcheck' function in Stata. However, we found no evidence that the linearity assumption had been violated. We also tested for interactions between sex and MRB; and each of the socioeconomic indicators and MRB, however, we found no strong evidence of any associations (p values

ranged from 0.047 for housing tenure to 0.827 for FSM). All analyses were conducted in Stata V.14.

## RESULTS
### Associations between total MRB and educational outcomes

In the ALSPAC cohort at age 16 years, we found that the total number of MRBs engaged in was strongly associated with lower educational attainment (see table 3). In the unadjusted models, for every additional risk behaviour a young person engaged in, the capped GCSE score decreased on average 9.17 points (95% CI −10.25 to −8.10, p<0.001), equivalent to a grade and a half in one GCSE examination. Similarly, the odds of attaining five or more A*–C grades were reduced by 18% (OR 0.82 (95% CI 0.79 to 0.85), p<0.001) for each additional risk behaviour.

The association between MRBs and capped GCSE score did not change markedly with the inclusion of sex and season of birth in the models. However, a more substantial reduction in the association was seen with the inclusion of the socioeconomic indicators (maternal education, parental social class, household income, housing tenure and FSM status). The final and fully adjusted model, which additionally included IQ at age 8 years and previous educational attainment, shows that for each additional MRB, the participant had a reduction in capped GCSE score of 6.31 points (95% CI −7.03 to −5.58, p<0.001) which is equivalent to just more than one grade in one GCSE examination. The mean number of risk behaviours engaged in was 3.24, which means that on average young people from this cohort had a reduced GCSE score of 20.44 points, associated with their risk behaviours. This is equivalent to a reduction of nearly three and a half grades in one GCSE examination, or one grade lower in three examinations.

The negative association between engagement in MRBs and gaining five or more GCSEs between A* and C, as demonstrated in the preliminary analysis, remained large in the adjusted models. Indeed, after adjusting for all the confounders, the magnitude of this effect increased and the odds of attaining five or more A*–C grades was reduced by 23% (OR 0.77 (95% CI 0.74 to 0.81), p<0.001) for each additional risk behaviour engaged in (although it should be noted that the confidence intervals do overlap with the previous model).

Each of the separate risk behaviours was negatively associated with educational attainment. Smoking had the largest deleterious association and those who smoked scored an average of 57.40 capped GCSE points less than those who did not smoke. This would be equivalent to getting an ungraded U classification instead of an A* in one GCSE examination, or put differently getting one grade lower in nine separate GCSE examinations. Smokers were also 70% less likely to gain five or more A*–C grades at GCSE compared with non-smokers at age 16 years. Not wearing a cycle helmet, hazardous alcohol use and physical inactivity were all negatively associated

**Table 2** Sample descriptive statistics

| | | Imputation sample 5401 % (SE) | Complete case sample 1617 n (%) |
|---|---|---|---|
| Educational outcomes | n=5401 | | |
| Capped GCSE score: mean (SE) | | 350 (1.01) | 376 (1.31) |
| Five or more A*–C grades at GCSE | | 74.4% (SE=0.6) | 1420 (87.8%) |
| Four or less A*–C grades at GCSE | | 25.6% (SE=0.6) | 197 (12.2%) |
| Exposure | n=2264 | | |
| MRB total: mean (SE) | | 3.24 (0.31) | 3.01 (0.05) |
| Sex | n=5401 | | |
| Female | | 55.5% (SE=0.7) | 975 (60.3%) |
| Male | | 44.5% (SE=0.7) | 642 (39.7% |
| Season of birth | n=5401 | | |
| Autumn | | 33.9% (SE=0.6) | 542 (33.5%) |
| Winter | | 14.8% (SE=0.5) | 222 (13.7%) |
| Spring | | 23.4% (SE=0.6) | 382 (23.6%) |
| Summer | | 27.9% (SE=0.6) | 471 (29.1%) |
| Maternal education | n=5214 | | |
| Degree | | 14.9% (SE=0.5) | 311 (19.2%) |
| A level | | 25.9% (SE=0.6) | 479 (29.6%) |
| O level | | 34.3% (SE=0.7) | 596 (36.9%) |
| <O level | | 21.5% (SE=0.6) | 231 (14.3%) |
| Parental socioeconomic position | n=4970 | | |
| Professional | | 14.1% (SE=0.5) | 291 (18.0%) |
| Managerial and technical | | 41.2% (SE=0.7) | 761 (47.1%) |
| Skilled non-man | | 23.6% (SE=0.6) | 398 (24.6%) |
| Skilled man, part or unskilled | | 13.2% (SE=0.5) | 167 (10.3%) |
| Housing tenure | n=5227 | | |
| Mortgage/owned | | 84.7% (SE=0.5) | 1.444 (89.3%) |
| Private rent | | 7.3% (SE=0.4) | 88 (5.4%) |
| Subsidised rent | | 8.6% (SE=0.4) | 85 (5.3%) |
| Income | n=4809 | | |
| High | | 20.86% (SE=0.6) | 364 (22.5%) |
| Mid high | | 21.96% (SE=0.6) | 424 (26.2%) |
| Middle | | 21.34% (SE=0.6) | 364 (22.5%) |
| Mid low | | 19.25% (SE=0.6) | 292 (18.1%) |
| Low | | 16.58% (SE=0.5) | 173 (10.7%) |
| Free school meals (FSMs) | n=5401 | | |
| Ever FSM | | 7.7% (SE=0.4) | 65 (4.0%) |
| Never FSM | | 92.3% (SE=0.4) | 1552 (96.0%) |
| Special educational needs (SENs) | n=5075 | | |
| No SEN | | 84.7% (SE=0.4) | 1522 (94.1%) |
| School action | | 6.0% (SE=0.4) | 68 (4.2%) |
| School action plus | | 1.9% (SE=0.2) | 18 (1.1%) |
| Statement of SEN | | 1.4% (SE=0.2) | 9 (0.6%) |
| Previous educational attainment/ability | | | |
| IQ at age 8: mean (SE) | n=4370 | 105 (0.24) | 109 (0.37) |
| KS2 educational attainment: mean (SE) | n=4753 | 830 (2.76) | 901 (3.94) |

GCSE, General Certificate of Secondary Education; KS2, key stage 2; MRB, multiple risk behaviour.

**Table 3** Associations between total MRB score and capped GCSE score and odds of gaining five or more GCSEs at grades A*–C

|  | Unadjusted (n=5401) | Model 1 (n=5401) | Model 2 (n=5401) | Model 3 (n=5401) |
|---|---|---|---|---|
| Capped GCSE | −9.17 (−10.25 to − 8.10) | −9.12 (−10.19 to −8.05) | −6.90 (−7.86 to −5.94) | −6.31 (−7.03 to 5.58) |
|  | P<0.001 | P<0.001 | P<0.001 | P<0.001 |
| Five A*–C | 0.82 (0.79 to 0.85) | 0.82 (0.79 to 0.85) | 0.84 (0.81 to 0.87) | 0.77 (0.74 to 0.81) |
|  | P<0.001 | P<0.001 | P<0.001 | P<0.001 |

Model 1: adjusted for sex and season of birth.
Model 2: adjusted for sex, season of birth, maternal education, parental social class, household income, housing tenure and FSM.
Model 3: adjusted for sex, season of birth, maternal education, parental social class, household income, housing tenure, FSM, IQ and previous educational attainment.
FSM, free school meal; GCSE, General Certificate of Secondary Education; MRB, multiple risk behaviour.

with educational attainment, however, the evidence was less compelling (with p values ranging from 0.007 to 0.934). Details of these analyses can be found in the online supplementary material.

## DISCUSSION

In our analysis of the ALSPAC cohort, adolescents with a greater number of health risk behaviours had poorer educational outcomes at age 16 years. While the fully adjusted models for both the complete case and the imputed datasets showed some attenuation in the estimates, the effect remained strong despite adjusting for a wide range of confounders. The fully adjusted model showed an associated reduction in capped GCSE score of 6.31 points, which would be equivalent to a reduction of more than one grade in one GCSE examination, for each additional MRB engaged in. A similarly adverse association with a young person's odds of gaining five or more GCSEs between A* and C was also observed with the odds of attaining five or more A*–C grades at GCSE reduced by 23% for each additional risk behaviour. We also corroborated associations between a wide range of individual MRB and educational outcomes at age 16 years.

This is the first longitudinal study in the UK to examine the association between multiple health risk behaviours and educational attainment. Current research in this area often considers single risk behaviours or small 'clusters' of risks, but none to our knowledge consider a large number of heterogeneous risks simultaneously. Owing to the scale and richness of the ALSPAC data, we have been able to control for a wide range of confounders, including socioeconomic measures, previous educational attainment and IQ.

However, there are several limitations to our analysis. First, the risk behaviours were all reduced to binary variables in order to construct the total MRB score, which leads to a loss of information. While each of the individual behaviours showed a negative association with educational attainment, these associations would perhaps be more robust if examined using a different classification of risk behaviour (eg, hazardous alcohol use and physical inactivity). However, we think it unlikely to

have an impact on the relationship between number of MRB and educational attainment. Second, by summing the risk behaviours to create a measure of total MRBs, we attribute equal weight to each behaviour, however, we have found no strong evidence for an alternative classification of MRB in terms of clusters of specific MRBs or latent classes. Third, there is considerable missing data on confounders and MRB which reduces power and may introduce bias. While our outcome variables, obtained through linkage, were observed for the majority of the participants, we opted to restrict our analyses (and our imputation) to the 5401 providing information on at least one risk behaviour. This subsample of ALSPAC was more likely to be female and to have higher IQ; and less likely to be from the lowest income quintile; to be living in privately or subsidised rental property; to have ever claimed eligibility for FSMs and have lower parental social class (p<0.001). The sample used is clearly not a random sample of those who enrolled, however, for bias to be present in our multivariable models would require the dependent variable (educational attainment) to be conditionally related to whether participants are included or excluded from this analysis. The pattern of positive associations observed between various factors and selection might lead us to anticipate collider bias in the form of attenuated estimates for MRB and educational attainment. However, by conditioning on gender, IQ, income, FSM eligibility, parental social class as well as MRB in the regression models, we propose that any residual association between educational attainment and selection should be minimal and hence so should any (attenuating) bias. Fourth, some of the MRBs were assessed using questionnaires, which have potential for recall bias and social desirability bias. Finally, we may not have controlled for all relevant confounders, for example, lone parent status and child maltreatment which are both associated with poorer educational outcomes were not included in the analysis. We have included multiple alternative confounders (parental social class, maternal education, housing tenure and claiming eligibility for FSMs), that are all themselves strongly associated with lone parent status and childhood maltreatment, however,

their omission may still be understood as a significant limitation of the study.

The findings from this research build on and are consistent with other studies which have shown positive associations between single risk behaviours or small numbers of similar risks and poorer educational attainment.[3 4 16–22 40–42] They also echo findings from a US study which found strong evidence of associations between higher educational attainment and membership to the most-healthy cluster of adolescents (although that study did not show the same dose–response relationship that we have found).[15]

Establishing the direction of the association between MRB and educational attainment would provide a valuable focus for future work in this area. Analysis of repeated measures of MRB at different time points throughout childhood and adolescence, would allow the exploration of any differences in the association between MRB and educational outcomes according to timing of MRB. Further research is also required to identify the early-life antecedents that are associated with adolescent MRB, as this would facilitate effective early intervention of those with the highest risk of engaging in harmful MRB.

## CONCLUSIONS

Our findings demonstrate for the first time that multiple health risk behaviours act as an important predictor of adverse educational outcomes, over and above a wide range of confounders including IQ at age 8 years and previous educational attainment. This finding could aid the identification and targeting of young people at risk of underachieving during their compulsory education. Further, by showing a dose–response relationship between the two, we have shown the importance of intervening in and reducing each and every risk behaviour. Preventing MRBs during adolescence could improve educational attainment and thereby directly and indirectly improve longer-term health outcomes.

**Acknowledgements** We are extremely grateful to all the families who took part in this study, the midwives for their help in recruiting them and the whole ALSPAC team, which includes interviewers, computer and laboratory technicians, clerical workers, research scientists, volunteers, managers, receptionists and nurses. The UK Medical Research Council and the Wellcome Trust (grant ref: 102215/2/13/2) and the University of Bristol provide core support for ALSPAC.

**Contributors** CW, RK, MH, RC and JH conceived and designed the study. CW and JH carried out the study, including acquiring and analysing the data. CW and JH interpreted the data. CW drafted the manuscript. RK, MH, RC and JH critiqued the manuscript for important intellectual content. All authors read and approved the final version of the manuscript. CW serves as guarantor. MH and RC are senior investigators for the National Institute for Health Research. CW and JH had full access to all of the data and can take responsibility for the integrity of the data and the accuracy of the data analysis. CW, RK, MH, RC and JH had access to the statistical outputs, reports and tables. All researchers are independent of the funding bodies.

**Funding** ALSPAC receives core support from the UK Medical Research Council and the Wellcome Trust (grant ref: 092731) and the University of Bristol. This work was undertaken with the support of the Centre for the Development and Evaluation of Complex Interventions for Public Health Improvement (DECIPHer), a UKCRC Public Health Research Centre of Excellence. Joint funding (MR/KO232331/1) from the British Heart Foundation, Cancer Research UK, Economic and Social Research Council, Medical Research Council, the Welsh Government and the Wellcome Trust, under the auspices of the UK Clinical Research Collaboration, is gratefully acknowledged.

**Competing interests** None declared.

**Patient consent** Not required.

**Ethics approval** Ethical approval for the study was obtained from the ALSPAC Ethics and Law Committee and local Research Ethics Committees.

**Provenance and peer review** Not commissioned; externally peer reviewed.

**Data sharing statement** The ALSPAC policy on data sharing is available at www.bristol.ac.uk/alspac. To discuss access to ALSPAC data, please contact the ALSPAC executive team on alspac-exec@bristol.ac.uk.

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
