## [Reviewer comments · BMJ Open]

ARTICLE DETAILS

TITLE (PROVISIONAL)	The effect of multiple risk behaviours in adolescence on educational attainment at age 16 years: a UK birth cohort study
AUTHORS	Wright, Caroline Kipping, Ruth Hickman, M Campbell, Rona Heron, Jon

VERSION 1 - REVIEW

REVIEWER	Karen Patte Department of Health Sciences, Brock University Catharines, Ontario, Canada None declared.
REVIEW RETURNED	05-Jan-2018

GENERAL COMMENTS	The paper is clearly and succinctly written, and adds to the literature demonstrating an association between risk behaviours and academic achievement. The key strengths of the study include a large longitudinal sample and the ability to control for wide range of confounders, including socio-economic measures and measures of prior educational attainment and ability. The failure or inability to control for SES and reverse causality is a prime limitation in the existing literature, making this study a valuable addition to the literature. Some minor comments are provided below. There are a number of limitations; however, the authors have explained these in the manuscript, and responded appropriately to best account for them. The primary limitation is the substantial amount of missing data (1,617 participants had complete data, from 13,954 enrolled in the cohort and 5,401 with educational attainment and risk factor data), which the authors do acknowledge. As expected, higher risk populations had a higher tendency of missing data. The complete case samples had better educational outcomes and lower risk scores in comparison to the imputation sample, presenting a potential for bias. Overall, my main concern would be to provide further rationale for the study and design, primarily the composite risk score of heterogeneous variables. Response to Previous Review: Based on the letter and revised manuscript, the authors responded appropriately to the three concerns and suggested revisions from the previous review. That is, they tested for nonlinearity (they found no evidence that the linearity assumptions were violated), now test for several
--

interaction effects (there was no evidence of interaction effects), and have added the sequence of regression models to the Methods section.

Introduction:

The Introduction section is clearly written and organized by remains quite brief in terms of the background, rationale, and existing literature base. For instance, the authors write “Less studies consider multiple risk behaviours simultaneously (2,22-24) fewer consider a larger number of heterogenous risk behaviours concurrently. (25-27).” (lines 44-47). It would be helpful to briefly describe overall what was found among the handful of studies that have been conducted on risk behaviours and academic achievement (e.g., inconsistencies?).

Also, a sentence or two explaining the rationale for examining multiple risk behaviours simultaneously (why is this important or what does this add, as opposed to examining risk factors separately?) and why they decided to include a heterogenous mixture of risk factors together. For the latter, while I know these behaviours do cluster together, some readers may argue looking at classes of behaviour (e.g., focusing on substance use related variables, or on health/movement behaviours [physical inactivity and screen use], etc.) may provide a clearer picture for targeting prevention efforts and hypothesizing on mechanisms.

Methods:

Authors acknowledge in the paper that the sample is not representative in terms of socioeconomic status.

Statistical approaches appear sound.

Multiple risk behaviours were assessed by self-report questionnaires. The potential for recall bias and social desirability bias are also present in this design, particularly around risk behaviours, which could be added to the limitations.

Thirteen distinct risk behaviours were included. The authors reference to a previous paper for further detail on the derivation of each behaviour, but some further information provided in the current manuscript would be useful. While risk behaviours often cluster together, it is difficult to understand the inclusion of certain variables. All variables may be considered markers of a higher risk lifestyle, but mechanisms for academic achievement are not obvious (e.g., car passenger risk, cycle helmet use, scooter risk). Further description somewhere in the paper would aid in understanding the motivation and reasoning behind the study, and seemingly would be simple to provide.

Scores of 0 to 13 were derived for each participant based on the 13 risk behaviours assessed. The authors recognize and explain that the reduction of each risk behaviour measure to a binary variable is a prime limitation.

The rationale for included selected confounders is not provided (e.g., month of birth).

Line 25-27, the “We additionally controlled for IQ score at age eight years and key stage two educational attainment in order to account for reverse causality...”. First, readers outside of the UK may not be familiar with “stage two educational attainment”.

	Second, rewording to this sentence to reflect that controlling for these variables may help rule out reverse causality, it cannot be said to fully discount the possibility. IQ scores are not perfectly aligned to academic achievement, and academic achievement could have declined in the time between stage two and the initiation of risk behaviours. Results: The argument and purpose of the paper is to examine multiple risk factors simultaneously, which assumes their effect is comparable and additive effect. However, the results indicate contrasting patterns of association between the separate risk behaviours. Not wearing a cycling helmet, hazardous alcohol use, and physical inactivity had no association. Again, a simple and brief explanation of why the particular risk behaviours were chosen would strengthen the paper (aside from the fact that data on them was available). Discussion: The results are explained in terms of the achievement impact of each additional risk behaviour engaged in and as a dose response effect. While understandable given the use of the MRB risk score and results, when considered in conjunction with the lack of individual effect of three of the risk behaviours included in this score, it could be argued that the impact depends on whether the additional risk behaviour is one of these three. The authors also acknowledge another limitation being that they attribute equal weight to each behaviour, which is not supported by their individual effects. On page 12 Line 29-39, the authors explain the limitation of the binary variables, and that the risk behaviours that had no individual effect may have had an impact if classified differently. They may want to also add that the variables showing an individual effect may differ if nonbinary. For example, differences in academic achievement appear probable between adolescents using cannabis once a week on the weekend with friends, when compared to their peers using cannabis every day, perhaps several times a day, and yet both are classified as cannabis users.
--	--

REVIEWER	Brittany Barker BC Centre on Substance Use, Canada None declared
REVIEW RETURNED	23-Jan-2018

GENERAL COMMENTS	Thank you for providing the opportunity to review this paper, which seeks to assess the cumulative impact of various risk factors on educational attainment (by age 16) among UK adolescents born in 1991-92. The analyses benefit from a rich data source, the ALSPAC, a larger birth cohort study. While this paper has many merits, a number of major concerns are present. Although the authors acknowledge on page 12 that attributing equal weight and summing the MRBs is a limitation of the study, I am unconvinced that an inability to find an alternative classification system justifies the current approach. Intuitively, not wearing a seatbelt the last time a participant was a passenger is qualitatively different than carrying a weapon or physically assaulting someone in the past year. Further, by equally attributing and combining such diverse
--

categories of risk behaviours the findings are less meaningful – what specific policy implications are the authors able to make? It is already well established that engaging in criminal activities, substance use and many other risk behaviours are negatively associated with educational attainment. Unless the authors are able to differentially weight the risk behaviours, I do not believe this paper in its current form offers a unique contribution to the literature. I have provided further detailed comments below, which I hope are helpful.

1. A major issue with this paper is the potential for selection bias introduced due to the large number of participants (>50%) that were excluded for missing MRB data (11,336 to 5,401) on page 8. Are the authors able to demonstrate that the ~6,000 excluded are not systematically different than the 5,401 retained? Age, sex, ethnicity, baseline household and family variables (with p-values) would be helpful.

Further, with regard to the 5,401 individuals retained in the analysis, are the authors able to provide more information on how the data were imputed? How much on average was missing from a participant?

2. There are several places where descriptive data is referenced without an associated value.

Page 5, line 28-39: comparison between sample and 1991 census data (e.g., more likely to be white and home owners, less likely to be from mothers of ethnic minorities or claim free school meals) is missing p-values and a reference to where census data was obtained.

3. Are the authors able to provide response rates for the two sources that make up the exposure variable – the self-completed clinic questionnaire and postal questionnaire on page 6.

4. The section on page 6 under Outcome measures is confusing for someone who is not familiar with the UK education system such as myself. What is “key stage four” and “A-levels” and why are they not assessed with their peers? I also am not sure why the reference to the UK law of compulsory education is important – did the law change? Similarly, I think it is important to note (if I’m reading this correctly) that UK grades range from A-G and U for a fail, as that differs from North America (A-F). Could the authors clarify this section for a broader readership?

5. Page 7, Possible Confounders section: the covariates are undefined, how is “parental social class” defined and how does it differ from household income? Also, season of birth is not listed as a covariate, although it is adjusted for in the three models. Can the authors justify why this is an appropriate covariate given its importance in the models? Lastly, are all the adolescents still living in the parental home at time of data collection? Are those in foster care or living with a relative still captured? Controlling for living situation and whether it was a 2 or single parent home also seem relevant.

Lastly, a limitation of this study that deserves noting is the absence of adjustment for childhood maltreatment - a strong and established risk factor for poor educational outcomes.

VERSION 1 – AUTHOR RESPONSE

Reviewer 1

1. It would be helpful to briefly describe overall what was found among the handful of studies that have been conducted on risk behaviours and academic achievement (e.g., inconsistencies?).

Author response: We have added the following text to the paper:

It has been shown that those with no or intermediate qualifications are more likely to engage in MRB compared to those who attended higher education. However, these studies relate to adult populations and have considered only a limited number or range of risk behaviours.

2. Also, a sentence or two explaining the rationale for examining multiple risk behaviours simultaneously (why is this important or what does this add, as opposed to examining risk factors separately?) and why they decided to include a heterogeneous mixture of risk factors together.

Author response: We have added the following text to the paper:

Further, evidence has shown that MRB are cumulatively associated with cardiovascular and all-cause mortality. 1,2 For example, having four healthy lifestyle behaviours is associated with a 4-fold difference in mortality compared to having none.¹ Similarly, a composite measure of MRB predicted preventable death, over and above the predictive value of single lifestyle behaviours.³ It has also been posited that many lifestyle behaviours are underpinned by the same neural circuitry, and as such when intervening on one behaviour, changes in other behaviours can be expected. 4 Finally, interventions responding to multiple risks may be more efficient and better value for money, providing potentially huge benefits for considering multiple, rather than single risk behaviours.

3. For the latter, while I know these behaviours do cluster together, some readers may argue looking at classes of behaviour (e.g., focusing on substance use related variables, or on health/movement behaviours [physical inactivity and screen use], etc.) may provide a clearer picture for targeting prevention efforts and hypothesizing on mechanisms.

Author response: Thank you for this comment, we had expected to find clustering of risk behaviours in another related piece of research we have done. However, much to our surprise, we did not find evidence of clustering, which is why we used an MRB measure is a simple count of the number of MRB. With that in mind, we have added the following text to the paper to clarify our position:

A previous analysis of these MRB data, using latent class analysis (LCA)⁵, found that the model with the best fit showed that the resulting classes simply varied according to the number of risk behaviours, rather than demonstrating distinct risk profiles based on classes of behaviours. Therefore, there is no justification for the inclusion of analyses based on classes of behaviour and an MRB measure need only comprise a count of the number of risk behaviours.

4. Multiple risk behaviours were assessed by self-report questionnaires. The potential for recall bias and social desirability bias are also present in this design, particularly around risk behaviours, which could be added to the limitations

Author response: Agreed, we have added the following text to the discussion:

Some of the MRB were assessed using questionnaires, which have potential for recall bias and social desirability bias.

5. The rationale for including selected confounders is not provided (e.g., month of birth)

Author response: The following statement should clarify the inclusion of season of birth and is added to the Possible Confounders section:

Season of birth has been shown to be an important predictor of educational attainment. In England, where the academic year runs from September 1st to August 31st, this means that children who are born in the autumn tend to outperform those who are born in the summer.⁶

6. Line 25-27, the “We additionally controlled for IQ score at age eight years and key stage two educational attainment in order to account for reverse causality...”. First, readers outside of the UK may not be familiar with “stage two educational attainment” Second, rewording to this sentence to reflect that controlling for these variables may help rule out reverse causality, it cannot be said to fully discount the possibility. IQ scores are not perfectly aligned to academic achievement, and academic achievement could have declined in the time between stage two and the initiation of risk behaviours.

Author response: The text has been changed as follows in the Possible Confounders section (change highlighted in red):

We additionally controlled for IQ score at age eight years and key stage two educational attainment in order to reduce the likelihood of reverse causality between early educational attainment performance and ability (reflected using IQ) and engagement in MRBs. Key stage 2 is the legal term which refers to the four years of schooling, when pupils are aged between 7 and 11 years. Key stage 2 culminates in standardized assessment tests (SATS), the results of which we have used here.

7. The results are explained in terms of the achievement impact of each additional risk behaviour engaged in and as a dose response effect. While understandable given the use of the MRB risk score and results, when considered in conjunction with the lack of individual effect of three of the risk behaviours included in this score, it could be argued that the impact depends on whether the additional risk behaviour is one of these three.

Author response: While we take your point, we wouldn't necessarily expect all individual behaviours to be equally negatively associated with educational outcomes. Indeed, all the MRB do show negative associations with educational attainment, however, strong evidence ($p < 0.001$) isn't present for 3 of the MRB. This is especially unsurprising for the measure of physical inactivity which is nearly ubiquitous among the sample (nearly 75% of those with data do not meet recommended exercise levels).

However, we feel our approach is justified given the rationale for exploring MRB (as outlined earlier in response 2) and given the potential benefits of developing interventions and policy recommendations that are focussed on reducing the number of adolescent MRB, irrespective of the type of risk behaviour.

Text changed to:

Not wearing a cycle helmet, hazardous alcohol use, and physical inactivity were all negatively associated with educational attainment, but the evidence was less compelling (with p values ranging from 0.007 to 0.934).

8. On page 12 Line 29-39, the authors explain the limitation of the binary variables, and that the risk behaviours that had no individual effect may have had an impact if classified differently. They may want to also add that the variables showing an individual effect may differ if nonbinary. For example, differences in academic achievement appear probable between adolescents using cannabis once a week on the weekend with friends, when compared to their peers using cannabis every day, perhaps several times a day, and yet both are classified as cannabis users.

Author response: While we agree with this point, there are always going to be different (and potentially better) ways to represent the data, we have chosen to construct the MRB measure using binary cut offs. We have done this to maintain consistency with our other related research on MRB5,7-9, and also so that it is easier to replicate using other cohort studies. Given that, we will leave it as it is.

Reviewer 2

1. Although the authors acknowledge on page 12 that attributing equal weight and summing the MRBs is a limitation of the study, I am unconvinced that an inability to find an alternative classification system justifies the current approach. Intuitively, not wearing a seatbelt the last time a participant was a passenger is qualitatively different than carrying a weapon or physically assaulting someone in the past year. Further, by equally attributing and combining such diverse categories of risk behaviours the findings are less meaningful – what specific policy implications are the authors able to make?

Author response: We agree, intuitively these behaviours are different, but our research suggests that it is not the type or clusters of particular risk behaviours (see Reviewer 1 Comment 3 response about our latent class analysis) that are important, rather it is the total number of risk behaviours engaged in.

We have shown in previous analyses that social patterning is inconsistent for individual risks⁸, but strongly related to total MRB score and that the number of MRBs is strongly related to key health and social outcomes at age 18 years, including dependent/harmful alcohol use, obesity, NEET, anxiety, depression, being in trouble with police, and problem gambling⁷.

Using this evidence, we hope we will be able to develop appropriate interventions and policy recommendations that are focussed on reducing the number of adolescent MRB, irrespective of the type of risk behaviour and given our findings to date we suggest this approach could have beneficial effects.

2. It is already well established that engaging in criminal activities, substance use and many other risk behaviours are negatively associated with educational attainment. Unless the authors are able to differentially weight the risk behaviours, I do not believe this paper in its current form offers a unique contribution to the literature. I have provided further detailed comments below, which I hope are helpful.

Author response: We agree, it is already established that engagement in specific risk behaviours is associated with poorer educational attainment. However, as already explained our previous latent class analyses provide no justification for doing anything other than having a total MRB score. What this paper addresses, making a new contribution to the literature, is to explore whether a larger number of risk behaviours is associated with poorer educational attainment.

3. A major issue with this paper is the potential for selection bias introduced due to the large number of participants (>50%) that were excluded for missing MRB data (11,336 to 5,401) on page 8. Are the authors able to demonstrate that the ~6,000 excluded are not systematically different than the 5,401 retained? Age, sex, ethnicity, baseline household and family variables (with p-values) would be helpful.

Author response: the 5,401 are more likely to have female gender and higher IQ; and less likely to be from the lowest income quintile; to be living in privately or subsidised rental property; to have ever claimed eligibility for free school meals; and come from the lower parental social class ($p < 0.001$). We have added this as a limitation in the discussion section.

4. Further, with regard to the 5,401 individuals retained in the analysis, are the authors able to provide more information on how the data were imputed? How much on average was missing from a participant?

Author response: Yes, no problem, I have put additional information re: the multiple imputation into the paper as follows:

All variables used in the analyses including all measures of MRB, educational attainment and potential confounders are included in the imputation model, along with many other measures of the exposures, outcomes and confounders that had been collected at different time points. These are included in the imputation routine as auxiliary variables to reduce bias by improving the precision of the imputation model. Monte Carlo errors were used to compare the results obtained when imputing 25, 100, 250, and 500 data sets. Imputed results shown have been pooled across the 500 data sets. Among this sample there was an overall average missingness of 18.23%.

5. There are several places where descriptive data is referenced without an associated value. Page 5, line 28-39: comparison between sample and 1991 census data (e.g., more likely to be white and home owners, less likely to from mothers of ethnic minorities or claim free school meals) is missing p-values and a reference to where census data was obtained.

Author response: We have changed the text to read as follows:

As Boyd et al. (2013) observe the ALSPAC 'enrolled sample' are more likely to be White (OR= 3.85 [95%CI: 3.50–4.24] $p < 0.001$) and less likely to be eligible for free school meals (OR= 0.46 [95%CI: 0.43–0.50] $p < 0.001$) than the NPD KS4 GME national sample and 'recent responders' to ALSPAC are more likely to be: female (OR= 1.88 [95%CI: 1.74–2.03] $p < 0.001$), White (OR=1.34 [95%CI: 1.10–1.62] $p = 0.004$) and less likely to be eligible for free school meals (OR= 0.51 [95%CI: 0.44–0.60] $p < 0.001$).

6. Are the authors able to provide response rates for the two sources that make up the exposure variable – the self-completed clinic questionnaire and postal questionnaire on page 6.

Author response: Yes, I have added the following text to the paper:

Of the 11,366 subjects with education outcome data, 8,398 (74.08%) were invited to the TF3 clinic and of those, 4,534 (53.99%) attended; and 8,017 (70.72%) were sent the questionnaire and of those, data was available for 4,052 (50.54%)

7. The section on page 6 under Outcome measures is confusing for someone who is not familiar with the UK education system such as myself. What is "key stage four" and "A-levels" and why are they not assessed with their peers? I also am not sure why the reference to the UK law of compulsory education is important – did the law change? Similarly, I think it is important to note (if I'm reading this correctly) that UK grades range from A-G and U for a fail, as that differs from North America (A-F). Could the authors clarify this section for a broader readership?

Author response: Thank you for this comment, I have changed the text to clarify, (changes in red) see below:

Pupils in England aged between 14 and 16 years complete compulsory schooling and take their GCSE (or equivalent) examinations, this period of their education is referred to as key stage four (KS4). At the time, the ALSPAC cohort were in school, UK law stated that pupils were to remain in compulsory education until the age of 16, so unlike A-Levels, which are taken two years later and are optional, GCSEs are one of very few occasions in a young person's life when their educational attainment is assessed along with most of their peers. Two outcomes relating to KS4 educational attainment were used in the analysis. Achieving five or more A*-C grades at GCSE was chosen because it is a minimum requirement for many post-16 education and training courses and as such

represents an important threshold for young people to exceed. The second outcome takes the individual scores for each GCSE, which are calculated as A*=58 through to G=16 and ungraded, U=0 (unlike in North America where grades range from A-F).

8. Possible Confounders section: the covariates are undefined, how is "parental social class" defined and how does it differ from household income?

Author response: I have included the following text, to clarify the differences between parental social class and the other socio-economic measures, see below:

parent's highest social class (professional, managerial and technical, skilled non-manual or manual, part or unskilled managerial employment), mother's highest educational level (degree, A-level, O-level/GCSE and less than O-level/GCSE), household income (divided into quintiles of high to low income), housing tenure during pregnancy (mortgage or own property, privately rented property or subsidised rental property) and claiming eligibility for a free school meal.

9. Also, season of birth is not listed as a covariate, although it is adjusted for in the three models. Can the authors justify why this is an appropriate covariate given its importance in the models?

Author response: I have included the following justification for the inclusion of season of birth as follows:

Season of birth has been shown to be an important predictor of educational attainment. In England, where the academic year runs from September 1st to August 31st, this means that children who are born in the autumn tend to outperform those who are born in the summer.⁶

10. Lastly, are all the adolescents still living in the parental home at time of data collection? Are those in foster care or living with a relative still captured?

Author response: Yes, looked after children are included in the analysis.

11. Controlling for living situation and whether it was a 2 or single parent home also seem relevant.

Author response: Yes, while we accept that having a lone parent is associated with poorer educational outcomes, we feel that we have included sufficient alternative confounders (parental social class, maternal education, housing tenure and claiming eligibility for free school meals), that are all themselves strongly associated with lone parent status. We are therefore happy to acknowledge this as a potential limitation in the discussion, however, we will add the caveat that the other confounders are strongly associated with lone parent status and as such, its omission will have a negligible effect.

12. Lastly, a limitation of this study that deserves noting is the absence of adjustment for childhood maltreatment - a strong and established risk factor for poor educational outcomes.

Author response: Again, while we accept that childhood maltreatment is associated with poorer educational outcomes, we feel we have already controlled for sufficient (overlapping) confounders but have added this as a limitation in the discussion section.

Text added to paper in response to comments 11 and 12 as follows:

Finally, we may not have controlled for all relevant confounders, for example, lone parent status and child maltreatment which are both be associated with poorer educational outcomes. However, we have included sufficient alternative confounders (parental social class, maternal education, housing tenure and claiming eligibility for free school meals), that are all themselves strongly associated with lone parent status and childhood maltreatment and as such, their omission will have a negligible effect.

1. Kvaavik E, Batty D, Ursin G, Huxley R, Gale C. Influence of Individual and Combined Health Behaviors on Total and Cause-Specific Mortality in Men and Women. Archives of Internal Medicine 2010; 170(8): 711-8.
2. McCullough ML, Patel AV, Kushi LH, et al. Following cancer prevention guidelines reduces risk of cancer, cardiovascular disease, and all-cause mortality. Cancer Epidemiology, Biomarkers & Prevention 2011; 20(6): 1089-97.
3. Tamakoshi A, Tamakoshi K, Lin Y, Yagyu K, Kikuchi S, Group JS. Healthy lifestyle and preventable death: findings from the Japan Collaborative Cohort (JACC) Study. Prev Med 2009; 48(5): 486-92.
4. Spring B, Schneider K, McFadden HG, et al. Make Better Choices (MBC): Study design of a randomized controlled trial testing optimal technology-supported change in multiple diet and physical activity risk behaviors. BMC Public Health 2010; 10(1): 586.
5. Heron J, Campbell R, Hickman M, Kipping RR. Latent class analysis of multiple risk behaviour in adolescents in the ALSPAC cohort. BMC Public Health under review.
6. Crawford C, Dearden L, Greaves E. The drivers of month-of-birth differences in children's cognitive and non-cognitive skills. Journal of the Royal Statistical Society Series A, (Statistics in Society) 2014; 177(4): 829-60.
7. Campbell R, Hickman M, Kipping RR, Smith M, Poulou T, Heron J. The association between multiple risk behaviours in adolescence and health and social outcomes in early adulthood: prospective cohort study. Int J Epidemiol under review.
8. Kipping RR, Smith M, Heron J, Hickman M, Campbell R. Multiple risk behaviour in adolescence and socio-economic status: findings from a UK birth cohort. European Journal of Public Health 2015; 25(1): 44-9.
9. MacArthur GJ, Smith MC, Melotti R, et al. Patterns of alcohol use and multiple risk behaviour by gender during early and late adolescence: the ALSPAC cohort. J Public Health 2012; 34(suppl_1): i20-i30.
10. Boyd A, Golding J, Macleod J, et al. Cohort Profile: the 'children of the 90s'-the index offspring of the Avon Longitudinal Study of Parents and Children. Int J Epidemiol 2013; 42(1): 111-27.

VERSION 2 – REVIEW

REVIEWER	Brittany Barker BC Centre on Substance Use, Canada None declared
REVIEW RETURNED	14-Apr-2018

GENERAL COMMENTS	I thank the authors for the thorough responses to the suggestions by myself and the other reviewer and feel the paper is much improved as a result. Particularly, the additions to the methods section about modeling procedures, missing data and justification for season of birth are very helpful and informative. However, there are a couple of points that need further clarification or addressing before I believe BMJ Open should move forward with the publication. 1. I am still not fully satisfied with the authors response to the concern over equal weighting of risk factors (comment 1) – something that was echoed by the other reviewer. The authors state on page 8: “A previous analysis of these MRB data, using latent class analysis (LCA)¹, found that the model with the best fit showed that the resulting classes simply varied according to the number of risk behaviours, rather than demonstrating distinct risk profiles based on classes of behaviours. Therefore, there is no
--

justification for the inclusion of analyses based on classes of behaviour and an MRB measure need only comprise a count of the number of risk behaviours.” The reference cited is under review with BMC Public Health and as it has not been properly vetted through the peer review process, I do not feel it serves as a justification to attribute physically assaulting someone vs. not wearing a seatbelt as the same risk to educational attainment. Perhaps the authors can include some of the other evidence they have provided as a theoretical justification for their decision? I defer to the Editors judgment on whether the authors have adequately responded to this concern; although I would add that maybe tempering the language, “therefore there is no justification,” as it is a bit of an overstatement.

2. Another significant concern I had with this study was the potential for selection bias given that more than half of the original sample was excluded for missing data and had asked if those excluded were systematically different to those included (comment 3). I thank the authors for adding the significant differences between the two groups as a limitation of their study, however, as suspected, those excluded exhibited multiple markers of vulnerability (lower IQ, lowest income quintile, eligible for free school meal, lower parental social class). As such, I think a sentence or two about this difference and the potential impact on the findings in the discussion is warranted.

3. Both the other reviewer and myself asked for clarity on what “key stage 2” (Reviewer 1, comment 6) or “key stage 4” meant (comment 7), which appears to be missing. Is it possible to include a footnote or a note in parentheses for those not familiar with UK education systems? I ended up looking it up; may I suggest after KS4 including: “(year 10-11, pupils 14-16 years old)” or something to that effect. I believe this addition would aid in the clarity for an international readership.

4. My last comment (#12) regarded the lack of controlling for childhood maltreatment as a known confounder for poor educational attainment. The authors acknowledged this but asserted they had provided “sufficient alternative confounders” (parental social class, maternal education, housing situation, eligibility for free school meal). The variables listed are markers of poverty, not childhood maltreatment, so I would suggest re-writing it and acknowledging that not being able to control for childhood maltreatment is a significant limitation of the study.

Additional minor notes:

1. A closer edit of the paper is needed; there are many long sentences that could be shortened and the tense changes from past to present (e.g., page 7, lines 12-18).

2. If possible, are the authors able to reduce the number of acronyms in the paper? I found it difficult to follow, particularly in the methods section. For example, page 7, line 15 “NPD KS4 GME” – NPD should be spelled out the first time and acronyms only used if they are repeated throughout the paper.

3. Lastly, the language could be tempered in places, using “past research has shown” or “a previous study found” (e.g., page 5, lines 9-11).

VERSION 2 – AUTHOR RESPONSE

1. I am still not fully satisfied with the authors response to the concern over equal weighting of risk factors (comment 1) – something that was echoed by the other reviewer. The authors state on page 8: “A previous analysis of these MRB data, using latent class analysis (LCA), found that the model with the best fit showed that the resulting classes simply varied according to the number of risk behaviours, rather than demonstrating distinct risk profiles based on classes of behaviours. Therefore, there is no justification for the inclusion of analyses based on classes of behaviour and an MRB measure need only comprise a count of the number of risk behaviours.”

The reference cited is under review with BMC Public Health and as it has not been properly vetted through the peer review process, I do not feel it serves as a justification to attribute physically assaulting someone vs. not wearing a seatbelt as the same risk to educational attainment. Perhaps the authors can include some of the other evidence they have provided as a theoretical justification for their decision? I defer to the Editors judgment on whether the authors have adequately responded to this concern; although I would add that maybe tempering the language, “therefore there is no justification,” as it is a bit of an overstatement.

Author response: Measures of participation in thirteen distinct risk behaviours at the ages of 15 and 16 years were derived from participants’ responses at two ALSPAC data collections during their late teens (see Table 1). The first was a self-completed questionnaire issued during a clinic attended at age 15 (median age 15 years and 5 months) and the second comprised responses to a postal questionnaire administered at age 16 (median age 16 years and 7 months). The MRB measure was informed by the work of Hurrelmann and Richter (2006), who present an integrative model of risk behaviour. They argue that while inadequate coping processes are ubiquitous during adolescence, these processes can result in very different health risk behaviours among young people. However, despite variations in presentation (e.g. physically hurting someone on purpose, verses not wearing a seat belt), risk behaviours reflect the very similar dimensions of either externalising; internalising; or evasive forms. 34 Lending weight to this position, previous analyses using this measure of MRB have shown that health risk behaviours are patterned according to gender. For example, antisocial and criminal behaviours, cannabis use and vehicle-related risk behaviours are more prevalent among males, whilst tobacco smoking, self-harm and physical inactivity are more prevalent among females. However, despite the gendered patterning of single risk behaviours, females and males engaged in a similar number of risk behaviours. 35 Similarly, another previous analysis showed that while the associations between individual risk behaviours and measures of socioeconomic status (parental social class, maternal education and income quintile), were highly variable, a more consistent relationship was established between the MRB measure and socioeconomic status. When compared with the highest social class, maternal education or income quintile, the odds of engaging in a greater number of multiple risk behaviours increased for each incremental decrease in social position. 23 Finally, an analysis of these MRB data, using latent class analysis (LCA),³⁶ showed that the resulting classes simply varied according to the number of risk behaviours, rather than demonstrating distinct risk profiles based on classes of behaviours. Having found no strong evidence for employing an alternative classification of MRB based on classes of behaviour the MRB measure comprises a count of the number of risk behaviours representing a breadth of domains of social and health risk including: sexual health, substance use, self-harm, vehicle related injury risk, criminal and antisocial behaviour (ASB), and physical inactivity. The derivation of each behaviour is discussed in more detail in an earlier paper. 23 For the purposes of the analyses reported here a total number of risk-behaviours score from 0 to 13 was derived for each participant.

2. Another significant concern I had with this study was the potential for selection bias given that more than half of the original sample was excluded for missing data and had asked if those excluded were systematically different to those included (comment 3).

I thank the authors for adding the significant differences between the two groups as a limitation of their study, however, as suspected, those excluded exhibited multiple markers of vulnerability (lower IQ, lowest income quintile, eligible for free school meal, lower parental social class). As such, I think a sentence or two about this difference and the potential impact on the findings in the discussion is warranted.

Author response: Text changed as follows:

Thirdly, there is considerable missing data on confounders and MRB which reduces power and may introduce bias. Whilst our outcome variables, obtained through linkage, were observed for the majority of the participants, we opted to restrict our analyses (and our imputation) to the 5,401 providing information on at least one risk behaviour. This subsample of ALSPAC were more likely to be female and to have higher IQ; and less likely to be from the lowest income quintile; to be living in privately or subsidised rental property; to have ever claimed eligibility for free school meals; and come from the lower parental social class ($p < 0.001$). The sample used is clearly not a random sample of those who enrolled however for bias to be present in our multivariable models would require the dependent variable (educational attainment) to be conditionally related to whether participants are included/excluded from this analysis. The pattern of positive associations observed between various factors and selection might lead us to anticipate collider bias in the form of attenuated estimates for MRB and educational attainment. However, by conditioning on gender, IQ, income, free school meal eligibility, parental social class as well as multiple risk behaviour in the regression models, we propose that any residual association between education and selection should be minimal and hence so should any (attenuating) bias.

3. Both the other reviewer and myself asked for clarity on what “key stage 2” (Reviewer 1, comment 6) or “key stage 4” meant (comment 7), which appears to be missing. Is it possible to include a footnote or a note in parentheses for those not familiar with UK education systems? I ended up looking it up; may I suggest after KS4 including: “(year 10-11, pupils 14-16 years old)” or something to that effect. I believe this addition would aid in the clarity for an international readership.

Author response: Pupils in England aged between 14 and 16 complete compulsory schooling and take their GCSE (or equivalent) examinations at key stage four (KS4).

Has been changed to:

Pupils in England aged between 14 and 16 years complete compulsory schooling during school years 10 and 11 and take their GCSE (or equivalent) examinations, this is referred to as key stage four (KS4).

Apologies, I thought I had already added this to the main text. I have now added the following text as track changes:

Key stage two is the legal term which refers to the four years of schooling, in maintained schools in England and Wales when pupils are aged between 7 and 11 years. Key stage two culminates in standardized assessment tests (SATs) at age ~11 years, the results of which have been used here.

4. My last comment (#12) regarded the lack of controlling for childhood maltreatment as a known confounder for poor educational attainment. The authors acknowledged this but asserted they had

provided “sufficient alternative confounders” (parental social class, maternal education, housing situation, eligibility for free school meal). The variables listed are markers of poverty, not childhood maltreatment, so I would suggest re-writing it and acknowledging that not being able to control for childhood maltreatment is a significant limitation of the study.

Author response: Text changed as follows:

We have included multiple alternative confounders (parental social class, maternal education, housing tenure and claiming eligibility for free school meals), that are all themselves strongly associated with lone parent status and childhood maltreatment, however their omission may still be understood as a significant limitation of the study.

5. A closer edit of the paper is needed; there are many long sentences that could be shortened and the tense changes from past to present (e.g., page 7, lines 12-18).

Author response: Done.

6. If possible, are the authors able to reduce the number of acronyms in the paper? I found it difficult to follow, particularly in the methods section. For example, page 7, line 15 “NPD KS4 GME” – NPD should be spelled out the first time and acronyms only used if they are repeated throughout the paper.

Author response: I have gone through and tried to reduce the number of acronyms so as to make this section clearer.

7. Lastly, the language could be tempered in places, using “past research has shown” or “a previous study found” (e.g., page 5, lines 9-11).

Author response: Done.

VERSION 3 – REVIEW

REVIEWER	Brittany Barker BC Centre on Substance Use, Canada None declared
REVIEW RETURNED	28-May-2018

GENERAL COMMENTS	I thank the authors for their attention to my previous concerns with the study and the comprehensive revisions. I think the manuscript is greatly improved and recommend this study for publication with BMJ Open.
--